# Is ChatGPT a Good Sentiment Analyzer?

**Zengzhi Wang  Qiming Xie  Yi Feng  Zixiang Ding  Zinong Yang  Rui Xia**[*]
School of Computer Science and Engineering,
Nanjing University of Science and Technology, China
zzwang@njust.edu.cn qmxie@njust.edu.cn rxia@njust.edu.cn
https://github.com/NUSTM/ChatGPT-Sentiment-Evaluation

## Abstract

Recently, ChatGPT has drawn great attention from both the research community and the public. We are particularly interested in whether it can serve as a universal sentiment analyzer. To this end, in this work, we provide a comprehensive evaluation of ChatGPT on the understanding of *opinions*, *sentiments*, and *emotions* contained in the text. Specifically, we evaluate it in three settings, including *standard* evaluation, *polarity shift* evaluation and *open-domain* evaluation. We conduct an evaluation on 7 representative sentiment analysis tasks covering 17 benchmark datasets and compare ChatGPT with fine-tuned BERT and corresponding state-of-the-art (SOTA) models on them. We also attempt several popular prompting techniques to elicit the ability further. Moreover, we conduct human evaluation and present some qualitative case studies to gain a deep comprehension of its sentiment analysis capabilities.

## 1 Introduction

Large language models (LLMs) have profoundly affected the whole NLP community with their amazing zero-shot ability on various NLP tasks (Brown et al., 2020; Rae et al., 2021; Chowdhery et al., 2022; Zhang et al., 2022a, *inter alia*). As a representative, ChatGPT[1] has appeared out of the blue via interacting with people conversationally. It can conduct fluent conversations with people, write code as well as poetry, solve mathematical problems (Frieder et al., 2023) and so on, which has attracted widespread public attention.

However, despite its huge success, we still know little about the capability boundaries, i.e., where it does well and fails. In this work, we are interested in how Chat-GPT performs on the sentiment analysis tasks, i.e., *can it understand the opinions, sentiments, and emotions contained in the text?* To answer this question, we conduct a comprehensive evaluation on 7 representative sentiment analysis tasks[2] and 17 benchmark datasets, which involves three different settings including *standard* evaluation, *polarity shift* evaluation and *open-domain* evaluation (refer to Figure 1). We compare ChatGPT with fine-tuned small

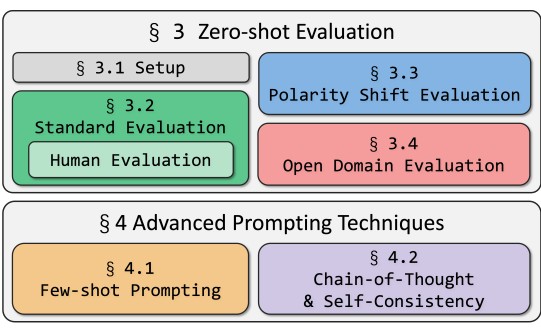

Table 1: The overview of our evaluation.

---

[*] Corresponding Author
[1] https://chat.openai.com/
[2] They are Sentiment Classification (SC), Aspect-Based Sentiment Classification (ABSC), End-to-End Aspect-Based Sentiment Analysis (E2E-ABSA), Comparative Sentences Identification (CSI), Comparative Element Extraction (CEE), Emotion Cause Extraction (ECE), and Emotion-Cause Pair Extraction (ECPE).

language models like BERT (Devlin et al., 2019) and corresponding SOTA models (if any) on each task for reference. We also attempt several popular prompting techniques, such as *chain-of-thought (CoT)* (Wei et al., 2022) and *self-consistency* (Wang et al., 2022), to induce the ability of ChatGPT. The main findings of our work are as follows:

❶ ChatGPT demonstrates impressive zero-shot capabilities in sentiment classification tasks such as SC, ABSC and CSI, and can rival fine-tuned BERT, although it still trails behind the domain-specific fully-supervised SOTA models (☞ § 3.2).

❹ ChatGPT seems less accurate on sentiment information extraction tasks like E2E-ABSA and CEE. We observe that ChatGPT can often make reasonable predictions but can not strictly match the dataset annotations. Our human evaluation finds that ChatGPT actually performs more desirable, not as poor as metrics indicate. (☞ § 3.2)

❸ Compared to fully-supervised strong baselines, ChatGPT demonstrates impressive textual emotion inference ability with significantly higher performance on ECE but lower performance on ECPE (☞ § 3.2).

❺ When coping with the *polarity shift* phenomenon (e.g., negation and speculation), a challenging problem in sentiment analysis, ChatGPT can make more accurate predictions than fine-tuned BERT. (☞ § 3.3)

❻ Compared to training domain-specific models, which typically perform poorly when generalized to unseen domains, ChatGPT demonstrates its powerful *open-domain* sentiment analysis ability in general, though its performance is quite limited in a few specific domains. (☞ § 3.4)

❼ Few-shot prompting (i.e., equipping with a few random examples in the input) can significantly improve performance across tasks and domains, surpassing fine-tuned BERT in some cases, though still inferior to SOTA models (☞ § 4.1). Applying CoT to the evaluated tasks does not yield gains but diminishes performance. In contrast, self-consistency reliably improves results (☞ § 4.2).

In summary, we think that, compared to training a specialized sentiment analysis system for each task, domain and dataset, **ChatGPT can already serve as a universal and well-behaved sentiment analyzer**.

## 2 Background and Related Work

### 2.1 Large Language Models

With the emergence of GPT-3 (Brown et al., 2020), Large language models (LLMs) were spotlighted. They typically have lots of model parameters and are trained on massive volumes of unstructured data at huge computational costs, including but not limited to Gopher (Rae et al., 2021), Megatron-Turing NLG 530B (Smith et al., 2022), LaMDA (Thoppilan et al., 2022), Chinchilla (Hoffmann et al., 2022), PaLM (Chowdhery et al., 2022), OPT (Zhang et al., 2022a), LLaMA (Touvron et al., 2023), and GPT-4 (OpenAI, 2023). As a result, given a simple task instruction, they are able to adapt directly to a new task in a training-free manner. In addition to the task instruction, the predictions will be more accurate and controllable if LLMs could be provided some demonstration examples, an ability known as *in-context learning* (Brown et al., 2020).

Lately, OpenAI has released ChatGPT, a chatbot fine-tuned from GPT-3.5 via reinforcement learning from human feedback (RLHF) (Christiano et al., 2017; Ouyang et al., 2022), drawing increasingly great attention. Next, researchers start exploring its abilities and limitations, testing it on various benchmarks (Gilson et al., 2022; Frieder et al., 2023; Guo et al., 2023; Jiao et al., 2023; Zhuo et al., 2023; Zhong et al., 2023; Ye et al., 2023; Laskar et al., 2023). For example, Bang et al. (2023) evaluate the multitask, multilingual, and multimodal aspects of ChatGPT, Wang et al. (2023a) conduct a robustness evaluation from the adversarial and out-of-domain perspective, and Borji (2023) summarizes 11 categories of failures towards ChatGPT. Related to our work, Zhong et al. (2023) analyze the language understanding

ability of ChatGPT on GLUE (Wang et al., 2018). In this work, we especially concentrate on analyzing its sentiment analysis ability, aiming to answer the question via a rigorous and comprehensive evaluation, i.e., *whether ChatGPT can be a good sentiment analyzer.*

## 2.2 Sentiment Analysis

Sentiment analysis seeks to identify people's *opinions*, *sentiments*, and *emotions* in the text, such as customer reviews, social media posts, and news articles (Liu et al., 2005; Liu, 2015). As one of the most active fields in Natural Language Processing (NLP), it has made rapid progress with the help of *deep learning* (Zhang et al., 2018; Yadav & Vishwakarma, 2020). Among the myriad of tasks associated with sentiment analysis, this paper is primarily concerned with 4 representative task categories, including (sentence-level) sentiment classification (SC), aspect-based sentiment analysis (ABSA), comparative opinion mining (COM), and emotion cause analysis (ECA). For ease of understanding, we will briefly introduce these tasks next. SC aims to identify the sentiment polarity of a given text, whether it is positive or negative. ABSA is designed to mine fine-grained aspect terms in the review and determine the sentiment polarity toward each aspect (Liu, 2012; Pontiki et al., 2014; Zhang et al., 2022b). We mainly focus on aspect-based sentiment classification (ABSC) and End-to-End Aspect Based Sentiment Analysis (E2E-ABSA) among many subtasks in ABSA. COM seeks to identify comparative sentences, extract the comparative elements, and obtain the corresponding comparative opinion tuples (Jindal & Liu, 2006; Liu et al., 2021). We mainly concentrate on comparative sentences identification (CSI) and comparative element extraction (CEE), i.e., extracting the tuple of (subject, object, comparative aspect, comparison type). The purpose of ECA is to extract the potential *cause clauses* given the emotion clause or extract the potential pair of *emotion clause* and *cause clause* in the text, which correspond to emotion cause extraction (ECE) (Gui et al., 2016) and emotion cause pair extraction (ECPE) (Xia & Ding, 2019), respectively.

In this paper, we are also concerned with two challenging problems in sentiment analysis: *polarity shift* and *open-domain* (Zong et al., 2021). Polarity shift refers to the linguistic phenomenon where the sentiment polarity (positive or negative) of a text shifts over time, context, or with respect to other texts (Li et al., 2010; Xia et al., 2016). Understanding sentiment *polarity shift* is crucial for building accurate sentiment analysis systems. As another challenging issue, *open-domain sentiment analysis* aims to understand the general sentiment of text regardless of the domain, whereas existing sentiment analysis systems typically focus on analyzing the sentiment of texts related to a particular domain (Cambria et al., 2012; Zhang et al., 2015; Luo et al., 2022). Addressing the above two issues is essential to building robust and effective sentiment analysis systems. In this work, we will examine whether ChatGPT can solve them.

## 3 Evaluation

In this section, we will first introduce the evaluation setup (§ 3.1) followed by *standard* evaluation (§ 3.2), *polarity shift* evaluation (§ 3.3) and *open-domain* evaluation (§ 3.4), as illustrated in Figure 1. As mentioned earlier, the tasks involved in our evaluation are SC, ABSC, E2E-ABSA, CSI, CEE, ECE, and ECPE.

### 3.1 Setup

**Comparison Systems.** We compare ChatGPT with the state-of-the-art (SOTA) (if any) models on end-tasks. Since SOTA models typically have some task-specific designs, we also provide the results of a commonly used baseline (e.g., fine-tuned BERT[3]) on each task for reference. For SC, we adopt the most common practice, i.e., using the final hidden representation of the [CLS] token as the sentence embedding and feeding it into a linear layer for classification. As for ABSC, we concatenate the review sentence and the aspect term via the special token [SEP] and classify the sentiment polarity based on the final

---

[3]All models use BERT-base-uncased version and are coupled with a linear layer if necessary.

hidden representation of `[CLS]`. We employ the joint tagging scheme (Li et al., 2019) to perform the E2E-ABSA task. For CSI, we report the performance of Multi-Stage$_{BERT}$ derived from (Liu et al., 2021) for reference. For CEE, given the complexity of modeling this task, we reformulate it into a text generation task based on `T5-Base` similar to GAS (Zhang et al., 2021), i.e., predicting the sequences of comparison tuples given the input review. We employ PAE-DGL (Ding et al., 2019) and ECPE-2D (Ding et al., 2020) as comparison models for ECE and ECPE, respectively. Unless otherwise specified, the above baseline models are rerun and repeated three times based on our evaluation settings.

**Usage of ChatGPT.** We mainly use ChatGPT with a specific version of `gpt-3.5-turbo-0301` for evaluation in this work, given its lower cost and improved performance (as stated in the OpenAI documentation[4]). We set the temperature to 0, making the outputs mostly deterministic for the identical inputs. Following Jiao et al. (2023), we ask ChatGPT to generate the task instruction for each task to elicit its ability to the corresponding task. For example, the prompt for E2E-ABSA is "`Given a review, extract the aspect term(s) and determine their corresponding sentiment polarity. Review: {sentence}`". Due to limited space, please refer to Table 9 and Appendix A.2 for complete prompts and prompts details, respectively. We report the zero-shot results of ChatGPT unless otherwise specified. We manually observe and record the predictions as the responses of ChatGPT do not always follow a certain pattern under the zero-shot setting.

**Evaluation Metrics.** We use accuracy and macro F1 score to evaluate sentiment classification tasks. We employ accuracy as the metric for CSI. For tasks involving elements extraction such as E2E-ABSA and CEE, we employ micro F1 score, i.e., a tuple is regarded as correct if and only if all elements inside it are exactly the same as the corresponding gold label. For ECE and ECPE, we compute the F1 score of cause clauses and emotion-cause clause pairs for evaluation, respectively.

| Task | Datasets | #Test | Metric | Fine-tuned | | Zero-shot |
| --- | --- | --- | --- | --- | --- | --- |
| | | | | **Baseline** | **SOTA** | **ChatGPT** |
| SC | SST-2 | 872 | Acc | 95.47[†] | **97.50**[α] | 93.12 |
| ABSC | 14-Restaurant | 1119 | Acc / F1 | 83.94[†] / 75.28[†] | 89.54 / **84.86**[β] | 83.85 / 70.57 |
| | 14-Laptop | 632 | Acc / F1 | 77.85[†] / 73.20[†] | **83.70 / 80.13**[γ] | 76.42 / 66.79 |
| E2E-ABSA | 14-Restaurant | 496 | F1 | 77.75[†] | **78.68**[δ] | 69.14 |
| | 14-Laptop | 339 | F1 | 66.05[†] | **70.32**[δ] | 49.11 |
| CSI | Camera | 661 | F1 | **93.04**[§] | - | 74.89 |
| CEE | Camera | 341 | F1 | **34.41**[♭] | - | 9.10 |
| ECE | Emotion Cause Dataset | 100 | F1 | 69.46[‡] | - | **74.01** |
| ECPE | Emotion Cause Dataset | 100 | F1 | **65.20**[♮] | - | 52.44 |

Table 2: Performance comparison among ChatGPT, fine-tuned baselines, and SOTA models on 9 datasets. #Test denotes the number of examples used for evaluation. † denotes the performance of fine-tuned BERT we implement. ‡ and ♮ denote the performance of PAE-DGL (Ding et al., 2019) and ECPE-2D (Ding et al., 2020) obtained by re-running experiments. § denotes the model performance of Multi-Stage$_{BERT}$ derived from Liu et al. (2021) while ♭ represents the results of our implemented GAS-Extraction-style baseline (Zhang et al., 2021). α, β, γ, and δ denote the results derived from T5-11B (Raffel et al., 2020), DPL (Zhang et al., 2022c), RILGNet (Li et al., 2022) and SyMux (Fei et al., 2022), respectively. The best results are in **bold**.

## 3.2 Standard Evaluation

In this part, we evaluate ChatGPT on 7 representative sentiment analysis tasks and report its results on related benchmark datasets.

**Datasets.** We choose SST-2 (Socher et al., 2013) as the testbed of SC. Since the test set of SST-2 is not public, we use its validation set for evaluation. We employ the SemEval

---

[4]`https://platform.openai.com/docs/models/gpt-3-5`

2014-ABSA Challenge Datasets (Pontiki et al., 2014) to evaluate the ability of ChatGPT to ABSA. For CSI and CEE, we employ the Camera dataset (Kessler & Kuhn, 2014; Liu et al., 2021). For ECE and ECPE, we adopt the Emotion Cause Dataset (Gui et al., 2016; Xia & Ding, 2019) and sample 100 examples from this. Except as noted above, we evaluate the remaining datasets on the full test set. The statistics are shown in the third column of Table 2.

**Results.** The comparison results are shown in Table 2. Overall, ChatGPT demonstrates highly competitive sentiment analysis performance compared with baseline models, albeit often being far inferior to SOTA models. Specifically, we observe that ChatGPT is on par with fine-tuned small language models (i.e., BERT) in sentiment classification tasks, despite being inferior to SOTA models. Secondly, when evaluated on E2E-ABSA, the performance of ChatGPT is indeed inferior to fine-tuned BERT, and the performance gap varies across domains. We speculate that the poorer performance on 14-Laptop is due to the presence of more proprietary terms and specific expressions in this domain. Thirdly, for the challenging COM tasks (i.e., CSI and CEE), which typically involve implicit expressions, although achieving reasonable performance on CSI, it exhibits extremely undesirable performance on CEE. These results are far from satisfactory compared with fine-tuned baselines. Finally, ChatGPT exhibits reasonably good emotion analysis ability. We find that ChatGPT can comprehend the given document thoroughly, for instance, being capable of identifying multiple reasons and extracting emotion clauses and cause clauses even when they are distant. We also observe that ChatGPT can make some reasonable predictions, whereas the corresponding annotations are not in the dataset.

**Human Evaluation.** In light of the poor performance on certain tasks, we naturally raise a question: *are the predictions of Chat-GPT truly unreasonable?* To acquire a more profound comprehension of the prediction results from ChatGPT, we conduct a human evaluation on E2E-ABSA and CEE owing to their unsatisfactory performance. Upon observation of the predicted results, Chat-GPT has made many plausible predictions.

| Task | Dataset | ChatGPT | +HumanEval |
|------|---------|---------|------------|
| E2E-ABSA | 14-Rest. | 69.14 | 83.86 |
|  | 14-Lap. | 49.11 | 72.77 |
| CEE | Camera | 9.10 | 51.28 |

Table 3: Human evaluation results on E2E-ABSA and CEE tasks.

However, these either did not exactly match the ground truth, or there are no corresponding annotations in the dataset, leading to a subpar performance on the exact-match evaluation. For E2E-ABSA, even though the predictions of ChatGPT are not accurate based on exact-match evaluation, it can still infer some highly reasonable aspect categories for the aspect terms thanks to its text generation paradigm. This also demonstrates its ability to identify implicit expressions to some extent. For instance, given the sentence "*Runs real quick.*", the ground truth is "(*Runs*, positive)" whereas the prediction of ChatGPT is "(*Speed*, positive)". For CEE, the predictions of ChatGPT express the same meaning as the ground truth but in an inconsistent form. As an example, the meaning expressed by ChatGPT is "*The SD800 is better than the SD700.*", whereas the ground truth meaning is "*The SD700 is worse than the SD800.*", where the "SD700" and "SD800" refer to the products being compared. From the perspective of sentiment analysis application, this is equally effective. Therefore, to align the predictions of ChatGPT with the annotation standard of existing datasets, we follow a few simple rules for human evaluation[5]:

☞ For any extra generated tuples, if they are reasonable but absent from the annotations, we will remove them from the prediction results. Otherwise, we will keep them.

☞ We also consider an aspect-sentiment or comparative opinion tuple correct if the boundary of aspect or entity is predicted incorrectly but unambiguously, and the predicted sentiment or preference is also correct.

☞ We also regard a prediction that paraphrases the ground truth to be correct, given the text generation paradigm.

---

[5]See Appendix A.4 for examples

As shown in Table 3, it is surprising but reasonable to observe that the zero-shot performance of ChatGPT is boosted by 19% (average) and 42% on E2E-ABSA and CEE, respectively, compared to the original results. These results suggest that ChatGPT's predictions are often reasonable, even though they may not strictly adhere to the dataset annotations, leading to the appearance of poorer performance.

**Case Study.** We also conduct the qualitative analysis for the predictions of ChatGPT. Due to the limited space, please refer to Appendix A.5.

### 3.3 Polarity Shift Evaluation

Comprehending the phenomenon of *polarity shift* in sentiment analysis is crucial for developing robust and reliable sentiment analysis systems. In this part, we evaluate the ability of ChatGPT to cope with the *polarity shift* problem. Specifically, we mainly focus on the situations of negation and speculation and consider two sentiment classification tasks, SC and ABSC.

**Datasets.** Since there are few datasets tailored to *polarity shift* for SC, we derive two subsets from SST-2 validation set using a heuristic rule for the evaluation of negation and speculation, namely SST-2-Negation and SST-2-Speculation. In short, it entails identifying whether a sentence contains any negation or speculation words. For instance, we assign a sentence to the negation evaluation subset if it includes the word "*never*". More details are provided in Appendix A.3. As for ABSC, we adopt the 14-Res-Negation, 14-Lap-Negation, 14-Res-Speculation, and 14-Lap-Speculation introduced by Moore & Barnes (2021), which are annotated for negation and speculation, respectively. The statistics are shown in Table 10.

**Baseline Details.** Generally, we fine-tune BERT on the original training set (e.g., SST-2) and evaluate on polarity-shifting test sets, e.g., SST-2-Negation and SST-2-Speculation.

**Results.** We conduct experiments on six evaluation datasets, and the comparison results are shown in Table 4. Compared to fine-tuned BERT, ChatGPT exhibits greater robustness in *polarity shift* scenarios. Essentially speaking, the *polarity shift* evaluation we conduct can be characterized as an *out-of-distribution* (OOD) evaluation scenario. Not surprisingly, we observe that fine-tuned BERT experiences varying degrees of performance degradation across datasets compared to standard evaluation results. In comparison, ChatGPT is more robust, especially on ABSC, where ChatGPT outperforms fine-tuned BERT by 10% in terms of average accuracy and 8% in terms of average F1 score. Furthermore, we also find that the speculation case in *polarity shift* appears more challenging than the negation case, as the results of the former is poorer.

| Task | Shifting Type | Dataset | Fine-tuned BERT | Zero-shot ChatGPT |
|------|---------------|---------|-----------------|-------------------|
| SC | Negation | SST-2-Neg. | **90.68** | **90.68** |
| | Speculation | SST-2-Spec. | **92.05** | **92.05** |
| ABSC | Negation | 14-Res-Neg. | 70.93
61.90 | **79.66**
**69.12** |
| | | 14-Lap-Neg. | 60.25
53.97 | **72.73**
**67.27** |
| | Speculation | 14-Res-Spec. | 64.29
60.53 | **77.01**
**68.45** |
| | | 14-Lap-Spec. | 40.86
39.40 | **47.47**
**46.96** |

Table 4: Performance comparison between ChatGPT and BERT on six datasets when dealing with negation and Speculation linguistic phenomena, measured by accuracy (top) and macro F1 score (bottom). The best results are in **bold**.

**Case Study.** We conduct qualitative analysis for the predictions of ChatGPT in the case of *polarity shift*. Refer to Appendix A.6 for details.

### 3.4 Open Domain Evaluation

Existing systems are typically trained on specific domains or datasets, leading to suboptimal generalization performance when dealing with unseen domains. However, an ideal sentiment analysis system could be applied to data from diverse domains. In this part,

| Model | Metric | Rest. | Lap. | Books | Cloth. | Hotel | Device | Service | Twitter | Finance | METS | Ave. |
|-------|--------|-------|------|-------|--------|-------|--------|---------|---------|---------|------|------|
| *Fine-tuned on the Rest. domain* | | | | | | | | | | | | |
| BERT | Acc. | 81.11 | **77.78** | 57.78 | 74.44 | **86.67** | 86.67 | 71.11 | 62.22 | 75.56 | 53.33 | 72.67 |
| | F1 | 74.99 | **70.60** | 41.91 | 55.00 | 77.59 | 85.35 | 67.91 | 54.11 | 62.75 | 47.06 | 61.14 |
| *Fine-tuned on the Lap. domain* | | | | | | | | | | | | |
| BERT | Acc. | **84.44** | 77.78 | 57.78 | 76.67 | **86.67** | 86.67 | 71.11 | 62.22 | 74.44 | 50.00 | 72.78 |
| | F1 | **78.76** | 72.84 | 42.84 | 56.21 | 76.94 | 88.92 | 67.59 | 56.16 | 55.59 | 37.56 | 60.78 |
| *Fine-tuned on the 9 out-of-domains each time* | | | | | | | | | | | | |
| BERT | Acc. | 80.00 | 76.67 | **62.22** | 76.67 | 85.56 | 94.44 | **81.11** | **70.00** | 31.11 | 38.89 | 69.67 |
| | F1 | 69.63 | 59.83 | 46.11 | **61.66** | 75.34 | 98.11 | **79.29** | **67.83** | 31.58 | 35.65 | 59.99 |
| *Fully-supvised results* | | | | | | | | | | | | |
| BERT | Acc. | 81.11 | 77.78 | 71.11 | 80.00 | 87.78 | 100.00 | 74.44 | 62.22 | 82.22 | 61.11 | 77.78 |
| | F1 | 74.99 | 72.84 | 57.17 | 58.15 | 77.98 | 100.00 | 62.69 | 60.99 | 79.07 | 58.53 | 67.64 |
| *Zero-shot results* | | | | | | | | | | | | |
| ChatGPT | Acc. | 83.33 | 73.33 | 60.00 | 70.00 | **86.67** | **96.67** | 76.67 | 66.67 | **86.67** | **76.67** | **77.67** |
| | F1 | 61.16 | 53.41 | **51.25** | 59.65 | **83.18** | **98.89** | 65.30 | 64.22 | **72.35** | **55.56** | **66.50** |

Table 5: Performance comparison between ChatGPT and fine-tuned BERT for ABSC task on open-domain evaluation. We also report the domain-specific fully-supervised results (in gray) of BERT for reference. The best results (except for fully-supervised results) are in **bold**.

| Model | Rest. | Lap. | Books | Cloth. | Hotel | Device | Service | Twitter | Finance | Mets-Cov | Ave. |
|-------|-------|------|-------|--------|-------|--------|---------|---------|---------|----------|------|
| *Fine-tuned on the Rest. domain* | | | | | | | | | | | |
| BERT | 76.55 | 43.57 | 38.35 | 29.57 | 64.07 | 50.74 | 27.01 | 1.67 | 7.74 | 3.27 | 34.25 |
| *Fine-tuned on the Lap. domain* | | | | | | | | | | | |
| BERT | 55.06 | 68.02 | 25.93 | 26.28 | 53.21 | **60.19** | 27.03 | 3.43 | 7.11 | 5.14 | 33.14 |
| *Fine-tuned on the 9 out-of-domains each time* | | | | | | | | | | | |
| BERT | 71.10 | **59.36** | **46.64** | **50.72** | **74.85** | 58.87 | **47.67** | **42.90** | 14.21 | **10.27** | **47.66** |
| *Fully-supvised results* | | | | | | | | | | | |
| BERT | 76.55 | 68.02 | 61.17 | 67.97 | 88.67 | 75.39 | 57.83 | 78.84 | 79.32 | 71.71 | 72.55 |
| *Zero-shot results* | | | | | | | | | | | |
| ChatGPT | **72.73** | 45.45 | 21.92 | 25.71 | 50.60 | 41.86 | 45.78 | 19.18 | **38.36** | 3.92 | 36.55 |
| + Human | 82.22 | 64.00 | 29.41 | 34.78 | 62.5 | 69.23 | 63.89 | 52.63 | 76.92 | 9.88 | 54.55 |

Table 6: Performance comparison between ChatGPT and BERT for E2E-ABSA task on the open-domain evaluation. We report the domain-specific fully-supervised results (in gray) of BERT for reference. We also report the human evaluation results ("+ Human") of ChatGPT for reference. The best results (except for fully-supervised results and human evaluation results) are in **bold**.

we evaluate the capability of ChatGPT to handle *open-domain* sentiment analysis tasks (i.e., ABSC and E2E-ABSA).

**Datasets.**    As there is currently no widely used *open-domain* evaluation dataset, we sample 30 examples from each domain of existing 10 ABSA datasets according to the original data distribution, resulting in a total of 300 samples both for ABSC and E2E-ABSA. The ten datasets involved are Restaurant (Pontiki et al., 2014), Laptop (Pontiki et al., 2014), Device (Hu & Liu, 2004), Service (Toprak et al., 2010), Books, Clothing, Hotel (Luo et al., 2022), Twitter (Dong et al., 2014), Financial News Headlines (Sinha et al., 2022), METS-CoV (Zhou et al., 2022), covering various domains such as restaurant reviews, product reviews, social media, finance, and medicine. Note that Books, Hotel, and Clothing are originally document-level ABSA datasets with hierarchical entity-aspect-sentiment annotations. We randomly sample 30 sentences from each dataset and only use the aspect-sentiment annotations.

**Baseline Details.**    To simulate the *open-domain* setting, we hold out some datasets, fine-tune BERT on the remaining datasets, and select checkpoints based on the mixture of the corresponding validation sets. Specifically, we set the following settings: (1) *single-source*: the model is trained on one dataset then evaluated on all datasets. Here, we choose Restaurant and Laptop as the testbed; (2) *multi-source*: the model is trained sequentially on nine datasets and then evaluated on the remaining one. Finally, we also fully-supervisedly fine-tune BERT and report the results for reference.

**Results.** In terms of ABSC, ChatGPT demonstrates a more compelling *open-domain* ability than BERT despite being fine-tuned on this task. As shown in Table 5, ChatGPT matches or even outperforms multi-domain fine-tuned BERT on 7 out of 10 domains in sentiment classification metrics (accuracy or macro-F1) while surpassing it by 8% in accuracy and 7% in F1 score on average across 10 datasets. It is worth mentioning that ChatGPT even performs comparably to full-supervised BERT, which shows its compelling generalization ability. Interestingly, fine-tuning on multiple domains does not necessarily lead to improved performance. For example, we observe that it results in a significant decrease in performance in certain datasets such as Finance and METS-Cov. Table 6 shows ChatGPT exhibits moderate performance on E2E-ABSA under the exact-match evaluation despite in the zero-shot manner. For example, it even beat BERT models on some domains (e.g., restaurant, service, and finance), which are fine-tuned on the nine domains.

Despite its success, we can observe that the performance of ChatGPT is quite poor in some domains, especially social media relevant domains (i.e., twitter, finance, METS-Cov), which suggests that improving performance on these domains remains challenging. It should be noted that due to the use of exact-match evaluation, the actual results of ChatGPT may not be as poor as they appear. Similarly, through our human evaluation (as introduced in § 3.2), we can observe that ChatGPT has achieved an average performance improvement of 18% across domains, indicating that many predictions are inherently reasonable. Overall, these results can demonstrate decent *open-domain* capabilities of ChatGPT, albeit with poor results in a few domains.

**Case Study.** We conduct qualitative analysis through four examples of ChatGPT on Books and METS-Cov, corresponding to the books and medicine domain, as shown in Figure 5. We also provided a detailed analysis in Appendix A.7.

## 4 Advanced Prompting Techniques

Given that ChatGPT still lags behind fine-tuned small language models (e.g., BERT) in some tasks and domains to a certain extent, we endeavor to seek help from some advanced prompting techniques to further elicit the capabilities of ChatGPT. Here, we adopt the ABSA tasks as the testbed.

### 4.1 Few-shot Prompting

We randomly select a few examples from the training dataset used for demonstration and concatenate them with the target input to prompt ChatGPT, a technique also known as *in-context learning* (Brown et al., 2020). We conduct few-shot prompting experiments on ABSC and ASPE with $k$ (i.e., 1, 3, 9 and 27) examples. To reduce the variance caused by the sampling of demonstration examples, we adopt three random seeds for sampling to conduct experiments and report the average performance. We compare the resulting performance with fully-supervised BERT and SOTA.

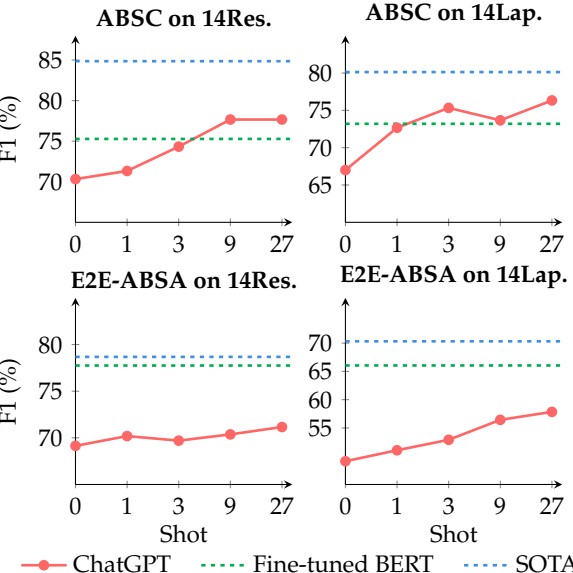

Table 7: Few-shot prompting results on ABSC and E2E-ABSA tasks.

**Results.** As presented in Figure 7, few-shot prompting can significantly improve the performance across tasks and datasets, even surpassing fine-tuned BERT in some cases. It

improves the classification performance by 7% and 10% F1 score for ABSC on 14-Restaurant and 14-Laptop, respectively, with 27 demonstration examples. We can also observe certain improvements on ASPE, although the improvement curve is relatively flat. We also provide a case study, as shown in Figure 1.

### 4.2 Chain-of-Thought and Self-Consistency

Although few-shot prompting clearly improves the performance on ABSC, the performance on E2E-ABSA still lags far behind fine-tuned BERT. We attempt more advanced techniques, i.e., *manual few-shot chain-of-thought (CoT) prompting* (Wei et al., 2022) and *self-consistency* (Wang et al., 2022) on this task, to further elicit the ability. More details are provided in the Appendix A.8.

**Results.** As shown in Table 8, we observe that equipping standard few-shot prompting with chain-of-thought does not bring the expected gains, but rather lead to a noticeable drop. This similar phenomenon was also observed in Ye & Durrett (2022) and Wang et al. (2022) but contrary to the observations in Zhong et al. (2023). We speculate that this may depend on the evaluation tasks. In contrast, self-consistency clearly improves the perfor-

| Prompting Methods | 14-Res. | 14-Lap. |
|---|---|---|
| Zero-shot prompting | 69.14 | 49.11 |
| Few-shot prompting (3 shot) | 69.70 | 52.90 |
| Few-shot prompting (9 shot) | 70.37 | 56.43 |
| Few-shot prompting (3 shot) + CoT | 67.24 | 46.28 |
| Few-shot prompting (9 shot) + CoT | 64.98 | 50.19 |
| 3-shot + Self-Consist. ($N = 5$) | 72.51 | 53.45 |
| 3-shot + Self-Consist. ($N = 10$) | 72.87 | 54.22 |
| 3-shot + Self-Consist. ($N = 15$) | **73.22** | **55.01** |
| 3-shot + CoT + Self-Consist. ($N = 5$) | 69.12 | 48.73 |
| 3-shot + CoT + Self-Consist. ($N = 10$) | 69.17 | 49.17 |
| 3-shot + CoT + Self-Consist. ($N = 15$) | 70.39 | 49.77 |
| Fine-tuned BERT | **77.75** | **66.05** |

Table 8: Results of advanced prompting techniques on E2E-ABSA. $N$ denotes the number of outputs sampled for the same input in the self-consistency technique.

mance of few-shot prompting, regardless of whether CoT is equipped, once again confirming the effectiveness of this technique (albeit at the cost of increased inference complexity). Regrettably, while effective, it is still inferior to fine-tuned BERT. Future work could explore more efficient prompting methods, such as retrieval-based ones (Liu et al., 2022; Shi et al., 2023, *inter alia*).

## 5 Conclusion

In this work, we evaluate ChatGPT on a range of test sets and evaluation scenarios and compare its performance to fine-tuned BERT, exploring its capacity boundaries in various sentiment analysis tasks. ChatGPT exhibits magnificent zero-shot sentiment analysis abilities (e.g., sentiment classification, comparative opinion mining and emotion cause analysis), even matching with fine-tuned BERT and SOTA models trained with labeled data in respective domains at times. Compared to fine-tuned BERT, ChatGPT can handle the *polarity shift* problem more effectively in sentiment analysis and exhibits good performance in *open-domain* scenarios. In addition, we also explore some popular prompting techniques to further induce the capability of ChatGPT. Through experiments, we validate the effectiveness of them on sentiment analysis tasks and provide our findings. We aspire to galvanize future research through our empirical insights in sentiment analysis, LLMs and beyond.

This work has several limitations: (1) Data Leakage: Difficulty in ensuring ChatGPT hasn't seen test data (Dodge et al., 2021; Golchin & Surdeanu, 2024; Balloccu et al., 2024; Xu et al., 2024; Sainz et al., 2024). (2) Prompt Design: Lack of extensive prompt experimentation assumes users prefer simplicity, yet improvements could showcase ChatGPT's adaptability. (3) Limited Evaluation: Focus on ChatGPT with limited comparison to other models due to resource constraints, narrowing insights into LLM strengths and weaknesses. Looking ahead, promising directions include (1) New Benchmarks: Developing real-world benchmarks and better evaluation methods to accurately assess LLMs (Zheng et al., 2023; Huang et al., 2024; Jimenez et al., 2024; Xie et al., 2024). (2) Implicit Sentiment Analysis: Enhancing

understanding and analysis of texts with implicit sentiments through comprehensive bench-marks. (3) Domain-Specific Enhancement: Improving ChatGPT's performance in specific domains, such as medicine (Singhal et al., 2023), math (Wang et al., 2023b; Azerbayev et al., 2024), code (Rozière et al., 2023) and low-resource languages (Dou et al., 2024; Nguyen et al., 2024), with targeted training for broader applicability. Due to space constraints, detailed discussions on limitations and future work directions are available in Appendix B.

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

# A  Appendix

## A.1  Closely Related Work

Prior to our work, there were some early studies that evaluated ChatGPT's performance on common NLP tasks, but they only covered basic sentiment classification tasks, such as SST-2 (Zhong et al., 2023). Our work, however, is dedicated to a more comprehensive and thorough evaluation for ChatGPT within the sentiment analysis field, representing, to our knowledge, the first of its kind in this field. After our work was completed (over a month later), a new study emerged, evaluating more tasks like ASTE and ASQP within ABSA and also providing few-shot experimental results and human evaluations (Zhang et al., 2023). However, this study did not evaluate critical issues in sentiment analysis such as sentiment polarity shift and open-domain evaluation, which our work does address. Although different works may have their own focuses, we believe they all contribute significantly to providing more comprehensive evaluations in this field.

## A.2  Prompts of ChatGPT

Following Jiao et al. (2023), we ask ChatGPT to generate the task instruction for each task to elicit its ability to the corresponding task. Taking the E2E-ABSA task as an example, our query is:

```
Please give me three concise prompts for eliciting your ability to perform
Aspect-Based Sentiment Analysis (i.e., extract the aspect terms and sentiment
polarity). There is no need to give examples and do not limit the prompts to a
specific product or domain.
```

Then, we examine the generated three prompts on a small-scale (e.g., 50 examples) example set driving from the corresponding training set. We select the best and most reasonable one[6] according to the results.[7] The final prompts adopted for each task are shown in Table 9. During the evaluation process, we feed a prompt and corresponding test example to ChatGPT and obtain a generated response. We manually observe and record the results as the responses do not follow a certain pattern.

## A.3  Preparation of Polarity Shift Evaluation Datasets

As previously mentioned, we drive SST-2-Neg and SST-2-Spec from SST-2 by detecting whether a sentence contains any negation or speculation words. The seed words adopted are shown in Table 11. And the statistics of involved datasets are shown in Table 10.

## A.4  Examples on Human Evaluation

The exact-match metric has limitations for evaluating generative models like ChatGPT since they can produce reasonable outputs not matched to references. To better characterize ChatGPT's capabilities despite this, we manually refine its outputs before comparing them to those of baselines. We acknowledge this may seem unfair compared to unrefined baselines. However, our goal is to account for the limitations of the exact-match, not to boost ChatGPT's results unfairly. To further illustrate the rules we use as more intuitive and easier to understand, we provide some examples from the E2E-ABSA task, as shown in Table 12.

## A.5  Case Study for Standard Evaluation

In this part, we conduct the qualitative analysis on ABSA tasks, COM tasks, and ECA tasks.

---

[6]When necessary, we would make minor adjustments to the prompts.

[7]We observe that different prompts have little effect on the performance. We also conducted three experiment repetitions and found minimal deviation in the results. Considering the cost of API calls, we only run the experiment once for the final evaluation unless otherwise specified.

| Task | Prompt |
|------|--------|
| SC | Given this text, what is the sentiment conveyed? Is it positive or negative? Text: {sentence} |
| ABSC | Sentence: {sentence} What is the sentiment polarity of the aspect {aspect} in this sentence? |
| E2E-ABSA | Given a review, extract the aspect term(s) and determine their corresponding sentiment polarity. Review: {sentence} |
| CSI | Does any comparison of products (including implicit products) exist in the product review: {sentence}? If so, outputs 'TRUE', else outputs 'FALSE'. |
| CEE | The following product review contains comparison of products (including implicit products): {sentence}. Extract the subject and object of comparison, tell me which aspect of products is being compared, and tell me if the author of the review thinks the subject is better or worse than or similar to or different from the object.\n If multiple comparisons exist, output multiple comparisons. |
| ECE | Document: {doc} \n Each line in the above document represents a clause and the number at the beginning of each line indicates the clause ID. Clauses expressing emotions are referred to as "emotion clause" and clauses causing emotions are referred to as "cause clauses". It has been identified that the clause with ID {emo_id}, {emotion clause} is an emotion clause, and the corresponding emotion keyword is {emotion}. Based on the above information, complete the following tasks: 1. Describe in one sentence the cause of the emotion clause with ID {emo_id}. 2. Based on the result of Task 1, output the ID of the cause clause that best fits the requirements. 3. According to the result of Task 2, match clauses with causality into pairs in the form "(emotion clause ID, cause clause ID)" and output all pairs as a set, such as (1,2),(3,4). Note: the emotion clause and the cause clause may be the same clause, and only the most obvious pairs need to be outputted. |
| ECPE | Document: {doc} \n Each line in the above document represents a clause and the number at the beginning of each line indicates the clause ID. Clauses expressing emotions are referred to as "emotion clause" and clauses causing emotions are referred to as "cause clauses". Based on the above information, complete the following tasks: 1. Describe the emotions and their corresponding causes contained in the document in one sentence. 2. Output the ID of the emotion clause in task 1, you only need to find the one with the strongest intensity. 3. For each emotion clause in task 2, find the corresponding cause clause and output the cause clause ID, you only need to find the most suitable one. 4. Match clauses with causality into pairs in the form "(emotion clause ID, cause clause ID)" and output all pairs as a set, such as (1,2),(3,4). Note: the emotion clause and the cause clause may be the same clause, and only the most obvious pairs need to be outputted. |

Table 9: The prompts used for prompting ChatGPT for each task. We manually design prompts for emotion cause analysis tasks (i.e., ECE and ECPE) due to the task complexity.

| Task | Dataset | #Test |
|------|---------|-------|
| SC | SST-2-Negation | 236 |
|    | SST-2-Speculation | 88 |
| ABSC | 14-Res-Negation | 1008 |
|      | 14-Res-Speculation | 448 |
|      | 14-Lap-Negation | 462 |
|      | 14-Lap-Speculation | 217 |

Table 10: The tasks and datasets involved in the polarity-shifting evaluation. #Test denotes the number of examples used for evaluation.

**Case Study on ABSA.** We conduct the qualitative analysis through two examples. Specifically, as shown in Figure 1, we present the results generated by ChatGPT for two test examples under zero-shot and few-shot settings, respectively. Given the example *"I did swap out the hard drive for a Samsung 830 SSD which I highly recommend"*, there are multiple aspect terms with different sentiment polarities in a sentence (e.g., the sentiment polarity

| Shifting Type | Seed Words |
|---|---|
| Negation | n't, no, not, never, neither, nor, unless, but, however, rather than, not yet, not only, nonetheless, despite, although, even though, in spite of, unlikely |
| Speculation | if, would, could, should, seems, might, maybe, whether, unless, even if, if only, can't believe, grant that, guessing, suspect, hope, wish, let's probably |

Table 11: Seed words used for deriving SST-2-Neg and SST-2-Spec from SST-2.

---

**Rule#1: For any extra generated tuples, if they are reasonable but absent from the annotations, we will remove them from the prediction results. Otherwise, we will keep them.**

Example#1
**Input:** It is super fast and has outstanding graphics .
**Output:**
    Aspect term: speed, graphics
    Sentiment polarity: positive, positive
**Ground Truth: [(graphics, positive)]**
**Refined Output:**
    **Aspect term: graphics**
    **Sentiment polarity: positive**

---

**Rule#2: We also consider an aspect-sentiment or comparative opinion tuple correct if the boundary of aspect or entity is predicted incorrectly but unambiguously, and the predicted sentiment or preference is also correct.**

Example#1
**Input:** the hardware problems have been so bad , i ca n't wait till it completely dies in 3 years , TOPS !
**Output:**
    Aspect term: hardware problems
    Sentiment polarity: negative
**Ground Truth: [(hardware, negative)]**
**Refined Output:**
    **Aspect term: hardware**
    **Sentiment polarity: negative**
Example#2
**Input:** And the fact that it comes with an i5 processor definitely speeds things up.
**Output:**
    Aspect term: processor
    Sentiment polarity: positive
**Ground Truth: [(i5 processor, positive)]**
**Refined Output:**
    **Aspect term: i5 processor**
    **Sentiment polarity: positive**

---

**Rule#3: We also regard a prediction that paraphrases the ground truth to be correct, given the text generation paradigm.**

Example#1
**Input:** Shipped very quickly and safely .
**Output:**
    Aspect term: Shipping
    Sentiment polarity: Positive
**Ground Truth: [(Shipped, positive)]**
**Refined Output:**
    **Aspect term: Shipped**
    **Sentiment polarity: Positive**
Example#2
**Input:** Runs real quick .
**Output:**
    Aspect term: Speed/Performance
    Sentiment polarity: Positive
**Ground Truth: [(Runs, positive)]**
**Refined Output:**
    **Aspect term: Runs**
    **Sentiment polarity: Positive**

---

Table 12: Examples on human evaluation. For simplicity, the task instruction is omitted.

of *"hard drive"* is neutral, and that of *"SSD"* is positive). We can observe that ChatGPT can not accurately identify the sentiment polarity of *"hard drive"* under the zero-shot setting. Similarly, in another test example *"I can say that I am fully satisfied with the performance that*

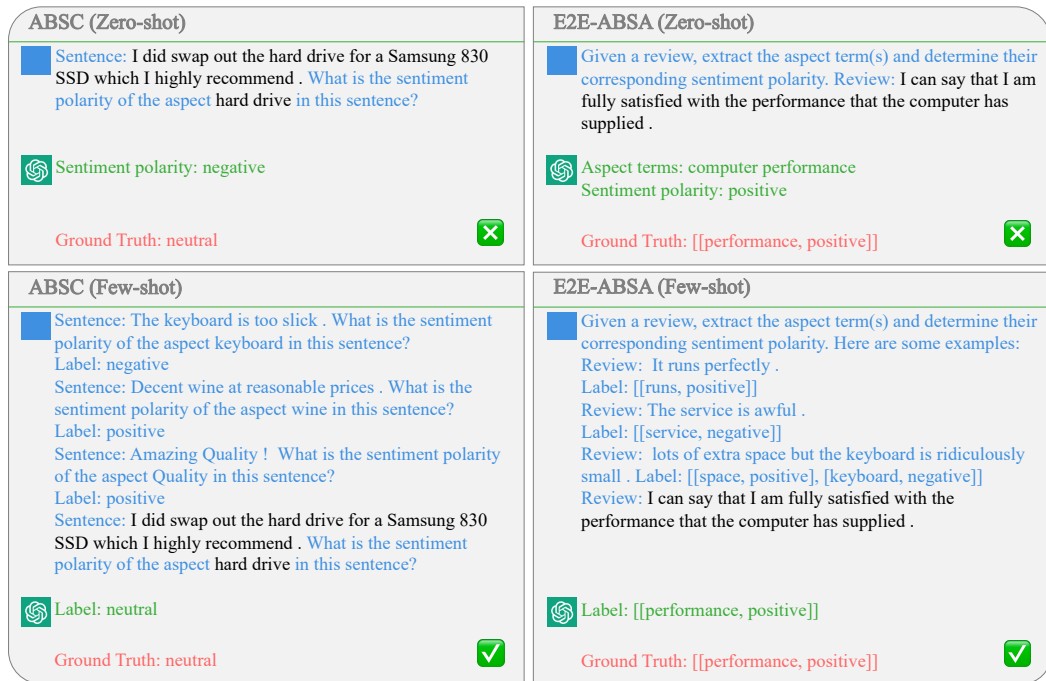

Figure 1: Case study for ChatGPT on ABSC and E2E-ABSA in zero-shot and few-shot settings. The text in blue, black, green and red denote the given prompts, the examples to be evaluated, the responses of ChatGPT and the ground truths, respectively.

*the computer has supplied."*, the aspect term extracted by ChatGPT is *"computer performance"*, which does not naturally exist in the sentence, indicating that ChatGPT may generate semantically reasonable aspect terms but without being aligned with the annotations in the dataset. However, under the few-shot setting (as introduced in § 4.1), after being equipped with a few demonstration examples, both of the above types of errors can be corrected by ChatGPT.

**Case Study on COM.**    We conduct qualitative analysis through two examples of ChatGPT in the case of CSI and CEE tasks, as shown in Figure 2. For the CSI task, it can be seen that ChatGPT is able to accurately identify explicit product comparison sentences. However, when the compared objects are implicit products, ChatGPT often considers the sentence not to be a comparison sentence, such as the sentence *"However, focus accuracy was not as impressive."* ChatGPT assumes that there are no explicitly mentioned products in the comment and therefore determines that it is not a comparison sentence. For CEE task, although ChatGPT is able to correctly identify comparison sentences and extract comparative elements, it tends to exhibit paraphrase phenomena when generating answers. For example, in the example sentence *"It seems to get less light to the sensors than my old 4MP A80."* the annotation indicates that the comparison subject is *"worse"* than the comparison object. However, when replying, ChatGPT expresses it as the comparison object is *"better"* than the comparison subject. This situation causes alignment issues between the generated answers and the annotations during automatic evaluation.

**Case Study on ECA.**    We also conduct qualitative analysis through two examples of ChatGPT in the case of ECE and ECPE tasks, as shown in Figure 3. It can be seen that ChatGPT can perfectly follow our given instructions to complete the task. Given the emotion of a document, ChatGPT can accurately analyze its corresponding cause, but the clause ID is not output as required (refer to the upper part of Figure 3). For the same document, without providing the emotion clause beforehand, we find that ChatGPT can still correctly identify it, which corresponds to the ECPE task (refer to the bottom part of Figure 3). Although the identified cause clause may not be the most obvious, the prediction is still reasonable.

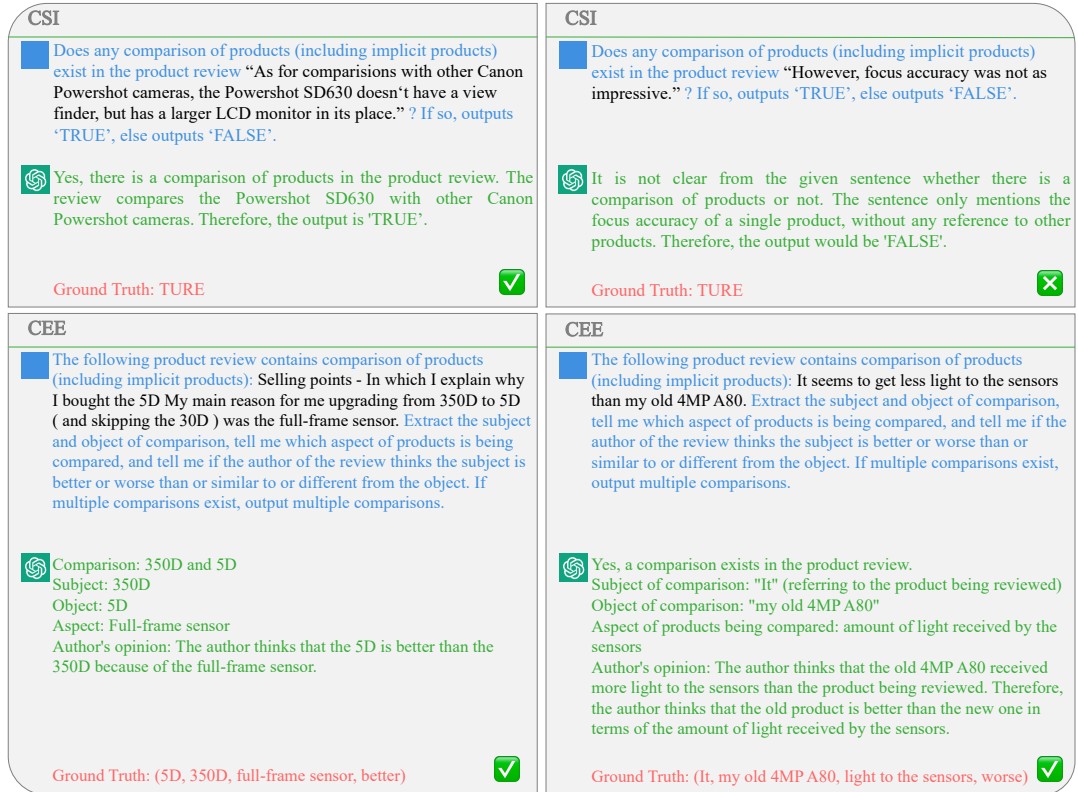

Figure 2: Case study for ChatGPT on CSI and CEE. The text in blue, black, green and red denote the given prompts, the examples to be evaluated, the responses of ChatGPT and the ground truths, respectively.

## A.6   Case Study for Polarity Shift Evaluation

We conduct qualitative analysis through four examples of ChatGPT in the case of *polarity shift* (i.e., negation and speculation), as shown in Figure 4. Observing the two examples of SC in the first row, it becomes apparent that ChatGPT can accurately determine the overall sentiment polarity of sentences accompanied by *polarity shift* due to the presence of negation and speculation expressions. Regarding the two examples in the second row of the ABSC task, when multiple aspect terms are listed in parallel and involve the linguistic phenomena such as negation and speculation (e.g., *"faster and sleeker looking"* and *"super fast and had outstanding graphics"*), ChatGPT struggles to identify their sentiment polarity accurately, leading to incorrect predictions.

## A.7   Case Study for Open-Domain Evaluation

We conduct qualitative analysis through four examples of ChatGPT on Books and METS-Cov, corresponding to the books and medicine domain, as shown in Figure 5. Regarding two examples in the first column, ChatGPT has difficulty accurately predicting sentiment in books domain since this domain usually contains unique expressions and plot descriptions that differ from typical product reviews. We also find that ChatGPT often generates reasonable aspect-sentiment pairs in the E2E-ABSA task, such as the *"(chapter creations, negative)"* (in the bottom left example). However, as we mentioned earlier, they are not originally annotated in the dataset, which is also an important reason affecting the performance of E2E-ABSA. As for the two examples in the second column, when dealing with a rare domain like medicine, ChatGPT can accurately determine the sentiment polarity of the given aspect term, but accurately extracting aspect-sentiment pairs remains a challenge.

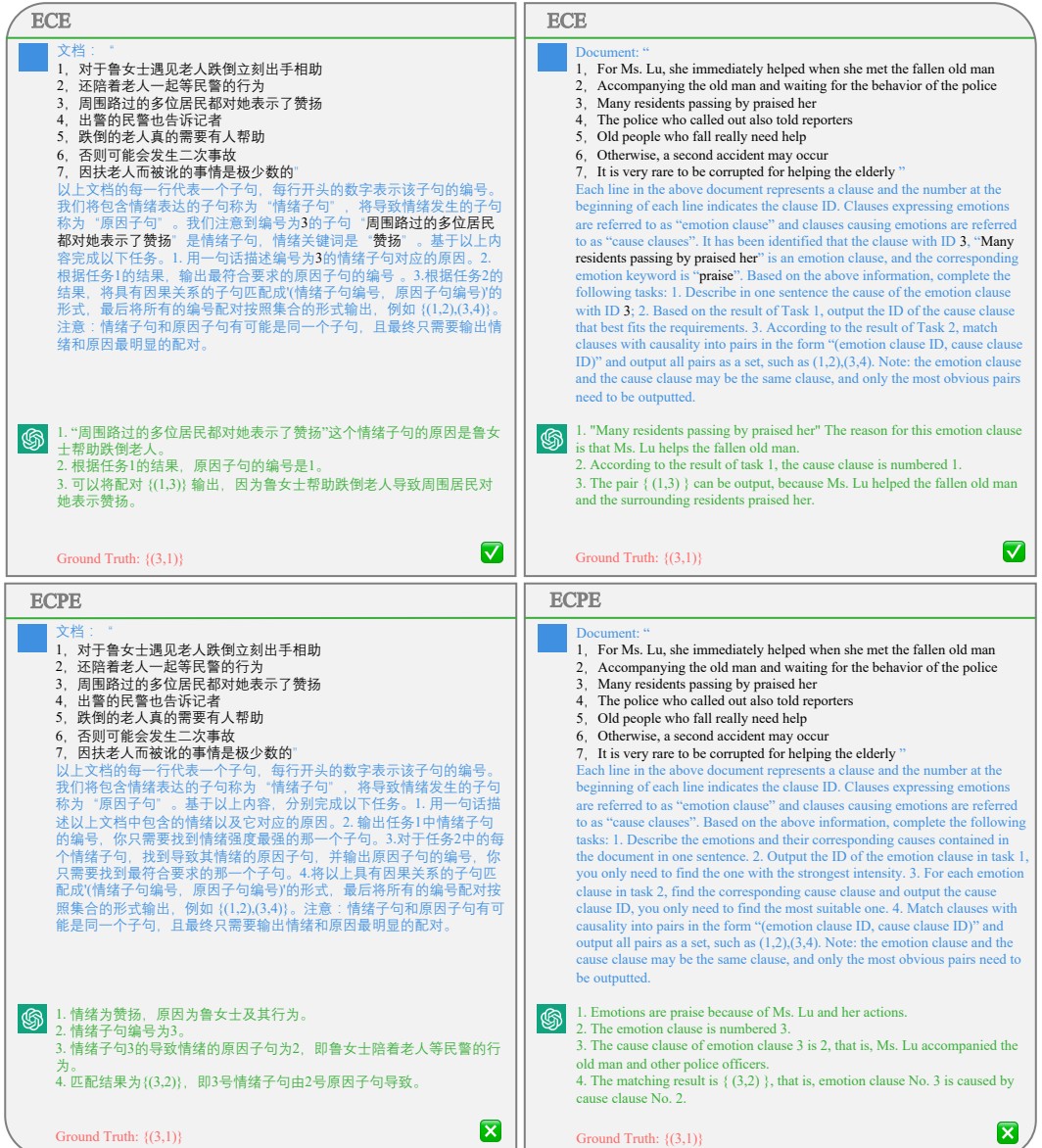

Figure 3: Case study for ChatGPT on ECE and ECPE in both Chinese (left) and English (right). The text in blue, black, green and red denote the given prompts, the examples to be evaluated, the responses of ChatGPT and the ground truths, respectively.

## A.8 Details for Chain-of-Thought and Self-Consistency Prompting

The chain-of-thought method augments each demonstration example in standard few-shot prompting with a chain of reasoning for the associated answer (Wei et al., 2022). We manually write CoT for randomly sampled examples. Self-consistency seeks to sample a diverse set of candidate outputs from LLMs and then aggregate the answers via a majority vote. We apply the temperature sampling with $T = 0.8$ as self-consistency is generally robust to sampling strategies (Wang et al., 2022). For the aggregation of answers, unlike the arithmetic reasoning task that typically has only one certain answer, the E2E-ABSA task we evaluate usually contains multiple aspect-sentiment tuples in an example. We adopt a heuristic approach by counting the frequency of each tuple in $N$ sampled predictions and filtering by setting a frequency threshold to obtain the final prediction. We can finely control

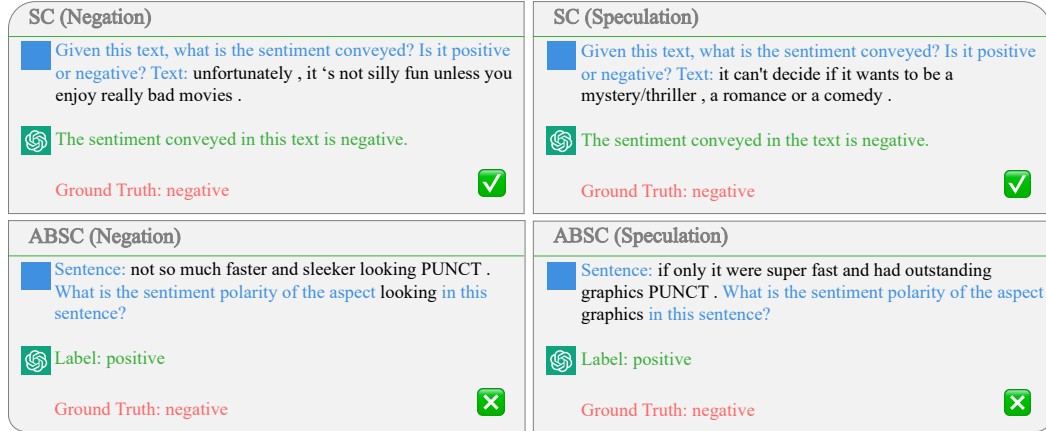

Figure 4: Case study for ChatGPT on SC and ABSC in case of the linguistic phenomena such as negation and speculation. The text in blue, black, green and red denote the given prompts, the examples to be evaluated, the responses of ChatGPT and the ground truths, respectively.

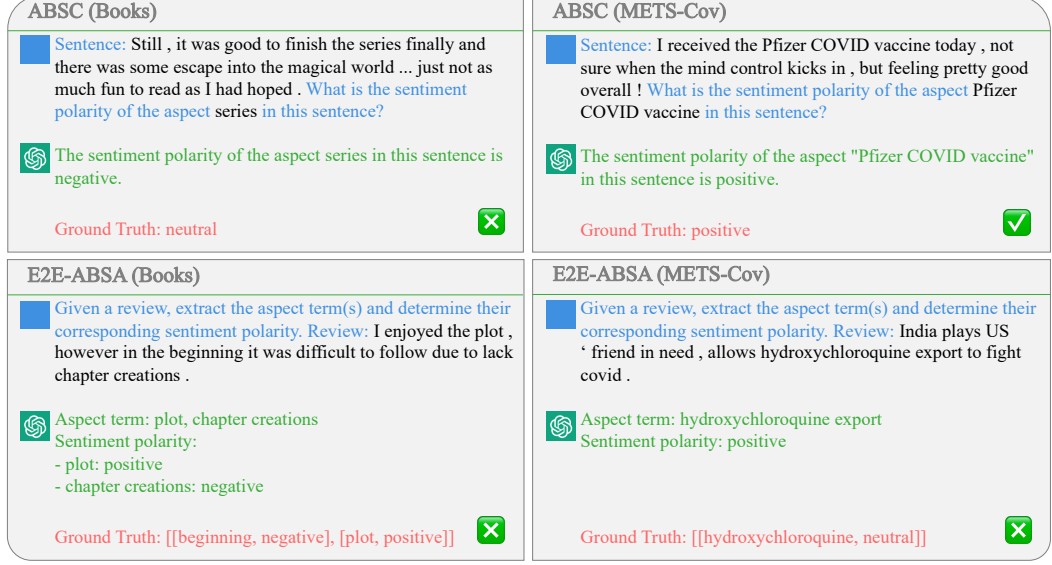

Figure 5: Case study for ChatGPT on ABSC and E2E-ABSA on books and medicine domains. The text in blue, black, green and red denote the given prompts, the examples to be evaluated, the responses of ChatGPT and the ground truths, respectively.

the answer aggregation by setting the threshold. In our experiments, we find that when $N = 15$, a threshold between 7 and 12 performs well.

## A.9 Other Evaluation Results

**Evaluation on text-davinci-003** Some readers might be curious about the performance of other powerful GPT-3.5 models in comparison to ChatGPT. To address this concern, we evaluate the powerful GPT-3.5 model, `text-davinci-003`, on some benchmarks. We carefully tune the evaluation to be as rigorous and controlled as possible, with temperature 0, top_p of 1, and 3 repeated runs to account for any variability (which is turned out to be negligible). As shown in Table 13, `text-davinci-003` achieves overall performance on par with ChatGPT.

| Task | Dataset | Metric | Baseline | SOTA | ChatGPT | text-devinci-003 |
|------|---------|--------|----------|------|---------|------------------|
| SC | SST-2 | Acc | 95.47 | **97.50** | 93.12 | 90.52 |
| ABSC | 14-Rest. | Acc / F1 | 83.94 / 75.28 | **89.54 / 84.86** | 83.85 / 70.57 | 82.19 / 71.74 |
| ABSC | 14-Lap. | Acc / F1 | 77.85 / 73.20 | **83.70 / 80.13** | 76.42 / 66.79 | 75.11 / 70.63 |
| E2E-ABSA | 14-Rest. | F1 | 77.75 | **78.68** | 69.14 | 65.06 |
| E2E-ABSA | 14-Lap. | F1 | 66.05 | **70.32** | 49.11 | 50.44 |

Table 13: Performance comparison among ChatGPT, text-davinci-003, fine-tuned baselines, and SOTA models on 5 datasets. Most results are derived from Table 2.

| Model | 14-Rest. | | 14-Lap. | |
|-------|----------|-----|---------|-----|
| | Implicit-split | All | Implicit-split | All |
| Fine-tuned BERT | **65.54** | **77.16** | **69.54** | **73.45** |
| ChatGPT | 56.31 | 69.72 | 52.68 | 65.92 |
| text-devinci-003 | 56.85 | 71.09 | 57.17 | 71.09 |

Table 14: Evaluation results on implicit sentiment analysis among fine-tuned BERT, ChatGPT and text-davinci-003.

**Evaluation on Implicit Sentiment Analysis**     As an interesting and challenging direction, we also explore the evaluation on implicit sentiment analysis. Following the dataset split of implicit sentiment analysis described in Li et al. (2021), we evaluate ChatGPT on the ABSC task and report BERT results (derived from Li et al. (2021)) as a reference. We also evaluate the performance of text-davinci-003. Similarly, we run 3 trials and report the average F1 over the implicit subset and the full ABSC dataset (we find that the variance is small). As shown in Table 14, we can observe that these large language models perform poorly on implicit sentiment analysis, although text-davinci-003 outperforms ChatGPT, both are weaker than fine-tuned BERT. These results suggest ample opportunities for future research.

## B   Limitations and Future Work

This work has several limitations as follows: (1) **Data leakage.** Currently, conducting rigorous evaluations for LLMs is extremely challenging. For example, it is difficult for us to determine whether the test set has been seen during the large-scale unsupervised pre-training, especially for models like ChatGPT, which are completely closed-source and can only be accessed through APIs (Dodge et al., 2021; Golchin & Surdeanu, 2024; Balloccu et al., 2024; Xu et al., 2024; Sainz et al., 2024, *inter alia*). Nevertheless, in this work, we still find some deficiencies of ChatGPT, such as its sentiment analysis performance in some domains (e.g., medicine and social media) that leaves much to be desired. (2) **Prompt design.** We do not conduct extensive prompt engineering, so there are likely better prompts to obtain better performance. Nevertheless, we believe that ordinary users usually do not do very delicate prompt designs when using LLMs. Therefore, if the ChatGPT can achieve sufficiently robust performance on arbitrary prompts, this would better demonstrate its capability. (3) **Limited evaluation.** Our evaluation is mainly conducted on ChatGPT, without including other equally powerful models. Although we have also supplemented other evaluation results in Appendix A.9, such as text-davinci-003. Unfortunately, such models are either completely closed-source and we do not have access to APIs, or we do not have enough GPUs to rigorously evaluate their performance due to their huge model parameters. However, as a representative of currently the most powerful models, evaluation on ChatGPT can also enable us to understand what LLMs currently do well and not well, thereby inspiring future research.

Beyond this work, we believe some promising future directions could include: (1) **New evaluation benchmarks.** We need to propose new and comprehensive benchmarks from real-world scenarios (Zheng et al., 2023; Huang et al., 2024; Jimenez et al., 2024; Xie et al., 2024, *inter alia*). Meanwhile, evaluation methods are also worth paying attention to. Due to the text generation paradigm, commonly used exact-match may not truly characterize the

model performance. In this paper, we adopt human evaluation to alleviate this issue. (2) **Implicit sentiment analysis.** Implicit expression is a very common linguistic phenomenon. For example, "I know real Indian food and this wasn't it" does not contain explicit opinion words. Moreover, accurate judgment often requires common sense or domain knowledge. Our experiments also confirm that large language models generally perform poorly on implicit sentiment analysis (See Appendix 14 for results). Meanwhile, constructing comprehensive benchmarks for implicit sentiment analysis could be a promising direction. (3) **Enhancing the model capabilities in specific domains.** As shown in Table 5 and Table 6, we can see that the performance of ChatGPT is not satisfactory on many domains (such as books and twitter). Therefore, in the future, we could improve the performance on certain domains through domain-specific training, as demonstrated by recent efforts (Singhal et al., 2023; Wang et al., 2023b; Azerbayev et al., 2024; Rozière et al., 2023; Dou et al., 2024; Nguyen et al., 2024, *inter alia*).

