# OpenReview forum: "Is ChatGPT a Good Sentiment Analyzer?"
_colmweb.org/COLM/2024/Conference — COLM_

### Official Review · Reviewer_Smbz · 2024-05-02

**Rating:** 7
**Confidence:** 3
**Ethics Flag:** 1

**Summary:**

The authors assess ChatGPT across various test sets and evaluation scenarios in order to delineate its potential in different sentiment analysis tasks. Specifically, they perform an assessment across 7 representative sentiment analysis tasks and 17 benchmark datasets, encompassing three distinct evaluation scenarios: standard evaluation, polarity shift evaluation, and open-domain evaluation. On those tasks, they compare ChatGPT against fine-tuned small language models such as BERT  along with any corresponding state-of-the-art models available for reference on each task. Finally, from these experiments, they conclude that ChatGPT can already function as a versatile and effective sentiment analyzer.

**Reasons To Accept:**

- They empirically prove that ChatGPT exhibits remarkable zero-shot sentiment analysis performance, encompassing tasks such as sentiment classification, comparative opinion mining, and emotion cause analysis. In certain instances, it even matches the performance of fine-tuned BERT and state-of-the-art models trained with labeled data in specific domains.
- Compared to fine-tuned BERT, ChatGPT demonstrates superior efficacy in addressing the polarity shift issue in sentiment analysis and performs admirably in open-domain contexts.
- They additionally explore popular prompting techniques to augment ChatGPT's capabilities further.
- The paper is well organized and easy to read.

**Reasons To Reject:**

The paper might have considered evaluating additional Large Language Models beyond GPT-3.5; however, I still believe it encompasses a satisfactory amount of work and quality to merit acceptance at COLM.

---

> ### Author Rebuttal · Authors · 2024-05-31
>
> Thank you for taking the time and effort to review our paper. Your recognition of our work is the greatest affirmation we could receive.

---

> > ### Comment · Reviewer_Smbz · 2024-06-05
> >
> > Thanks for the reply.

---

### Official Review · Reviewer_Tg6F · 2024-05-09

**Rating:** 6
**Confidence:** 5
**Ethics Flag:** 1

**Summary:**

The paper is an evaluation of the ability of GPT3.5 in different tasks of sentiment analysis. In particular de evaluation is interesting, and the conclusions related to the self-consistency prompt enhance the classification results. However, I miss a wider evaluation, since the aim of the paper is the evaluaation of the sentiment classification capacity of LLMs, so:

1. Why didn't the authors evaluate at least an additional LLM (Llama or Mistral for instance)?
2. Regarding the open domain evaluation, why didn't the authors use the SFU Review Corpus [1] that has 8 different opinion domains?

References:
[1] Taboada, M., Anthony, C., & Voll, K. D. (2006, May). Methods for Creating Semantic Orientation Dictionaries. In LREC (pp. 427-432).

**Questions To Authors:**

Why didn't the authors evaluate at least an additional LLM (Llama or Mistral for instance)?
Why didn't the authors use the SFU Review Corpus [1] that has 8 different opinion domains?
Why didn't the authors consider other language different than English?

References:
[1] Taboada, M., Anthony, C., & Voll, K. D. (2006, May). Methods for Creating Semantic Orientation Dictionaries. In LREC (pp. 427-432).

**Reasons To Accept:**

- I think that this evaluatino is useful in order to consider LLM as text classification methods.
- The wide evaluation of different sentiment analysis tasks.
- The  good results.

**Reasons To Reject:**

- The lack of assessing other LLMs.
- Tha lack of using more datasets, as the SFU Review corpus, and also datasets of microblogging genre.
- The lack of evaluation in other language than English, for instance the Spanish language.

---

> ### Author Rebuttal · Authors · 2024-05-31
>
> We greatly appreciate your time and efforts in reviewing our paper and offering insightful comments.
>
> We actually evaluated the performance of `text-davinci-003` in Tables 13 and 14 (in Appendix). For the language, the benchmark used for ECE and ECPE is in Chinese (refer to Fig. 3 for examples).
>
> We will update the results of other LLMs and benchmarks in our next version.

---

### Official Review · Reviewer_wJss · 2024-05-09

**Rating:** 6
**Confidence:** 4
**Ethics Flag:** 1

**Summary:**

This paper investigates ability of ChatGPT for various sentiment analysis tasks. The authors tried zero-shot and few-shot experiments with ChatGPT as well as some specific evaluations for samples with negations. The authors compared ChatGPT with BERT and SOTA baselines. The paper claims that ChatGPT can already serve as a universal and well-behaved sentiment analyzer.

In my opinion, the paper contributes to the thorough studies for sentiment analysis with the aim to evaluate ability of ChatGPT, while there exists limitations, such as the lack of consideration of other LLMs. In addition, I think the results are not surprising given the strong evidences of recent papers describing the ability of ChatGPT in other text classification and relation extraction tasks. I also would like to say that this paper is more likely about technical report because the analyses and claims are too broad.

**Reasons To Accept:**

- Thorough analyses with various subtasks of sentiment analysis (including E2E-ABSA): The authors provide different corpora and tasks in experiments, including major SST-2, Restaurant, Laptop datasets.
- The experimental setups and results seem scientifically valid
- The human analysis and case study are provided

**Reasons To Reject:**

- The findings of this paper are specific for ChatGPT, limiting generalizability for other LLMs. Given recent rapid trends in this field, using open-sourced LLMs such as Llama and Mistral would be interesting.
- Related to above, any findings can be obtained from this paper, but they may not be systematic results due to the limited generalizability.
- The authors use BERT as a baseline. However, I believe at least RoBERTa is a suitable baseline among fine-tuned small language models. It is also known that RoBERTa is a strong baseline in sentiment analysis tasks [1].
- The analyses in this paper are too broad in my opinion to convey the major claim that "ChatGPT can already serve as a universal and well-behaved sentiment analyzer".  Also, I think the results are not surprising given the strong evidences of recent papers describing the ability of ChatGPT in other text classification and relation extraction tasks. I also would like to say that this paper is more likely about technical report because the analyses and claims are too broad.

[1] Does syntax matter? A strong baseline for Aspect-based Sentiment Analysis with RoBERTa, Junqi Dai, Hang Yan, Tianxiang Sun, Pengfei Liu, Xipeng Qiu, NAACL 2021.

---

> ### Author Rebuttal · Authors · 2024-05-31
>
> Thanks for your insightful comments. We will update the results of RoBERTa and other LLMs in our next version.

---

> > ### Comment · Reviewer_wJss · 2024-06-04
> > **Thank you for your response**
> >
> > Thank you. I am happy that the results of RoBERTa and other LLMs will be described in the revised paper. The response does not make my evaluation of this paper change, so I would like to keep the score.

---

### Official Review · Reviewer_gr5x · 2024-05-21

**Rating:** 8
**Confidence:** 4
**Ethics Flag:** 1

**Summary:**

The paper is devoted to testing ChatGPT in various sentiment analysis tasks.Evaluation is done  on 7 sentiment analysis tasks covering 17 benchmark datasets. ChatGPT is compared with fine-tuned BERT and corresponding state-of-the-art approaches. The tasks include:  Sentiment Classification (SC), Aspect-Based Sentiment Classification (ABSC), End-to-End Aspect-Based Sentiment Analysis (E2E-ABSA), Comparative Sentences Identification (CSI), Comparative Element Extraction (CEE), Emotion Cause Extraction (ECE), and Emotion-Cause Pair Extraction (ECPE).

**Questions To Authors:**

No

**Reasons To Accept:**

1) ChatGPT is evaluated in diverse sentiment analysis tasks. Interesting conclusions are obtained.

2) Human evaluation is made to analyse low results of ChatGPT in some tasks.

3) Advanced prompting techniques such as few-shot prompting, chain-of-thought promptind are also studied.

4) The detailed description of expriments is well-presented in appendices.

**Reasons To Reject:**

1. Among all current exisiting LLM, only ChatGPT is tested.
2. All results are obtained for English, the conclusions can be quite different for other languages.

---

> ### Author Rebuttal · Authors · 2024-05-31
>
> Thanks for your time and efforts in reviewing our paper. Moreover, we sincerely appreciate your recognition of our work.

---

### Decision · Program_Chairs · 2024-07-10

**Decision:**

Accept

**Comment:**

The paper offers a comprehensive evaluation of ChatGPT's performance across a wide range of sentiment analysis tasks, including Sentiment Classification (SC), Aspect-Based Sentiment Classification (ABSC), and Emotion-Cause Pair Extraction (ECPE), among others. The experiments are well-detailed and include advanced prompting techniques such as few-shot prompting and chain-of-thought prompting. The inclusion of human evaluation to analyze low-performing tasks adds depth to the analysis. The experimental setups are scientifically valid, and the results are well-documented in the appendices. The study's thoroughness in testing multiple sentiment analysis subtasks using diverse datasets, including major ones like SST-2, Restaurant, and Laptop datasets, provides valuable insights into the capabilities of ChatGPT as a sentiment analyzer.

The study's primary limitation is its exclusive focus on ChatGPT, neglecting other large language models (LLMs) such as Llama and Mistral, which could provide a more comprehensive evaluation of LLM capabilities in sentiment analysis. The analyses are criticized for being too broad, making the claims that ChatGPT can serve as a universal sentiment analyzer seem overstated. Additionally, the paper does not explore results in languages other than English, limiting the generalizability of its conclusions. The use of BERT as a baseline, instead of a more robust model like RoBERTa, further limits the comparative analysis. The absence of evaluations on diverse datasets like the SFU Review Corpus and microblogging genres also restricts the study's applicability. Overall, the paper reads more like a technical report due to its broad and somewhat unsystematic analyses.

[comments from the PCs] As much as possible, please follow up on the AC recommendation to include an open-weights LLM, such as Llama or Mistral.